# Factors Affecting Road Tunnel Construction Accidents in China Based on Grounded Theory and DEMATEL

**DOI:** 10.3390/ijerph192416677

**Published:** 2022-12-12

**Authors:** Yanqun Yang, Yu Wang, Said M. Easa, Xiaobo Yan

**Affiliations:** 1College of Civil Engineering, Fuzhou University, Fuzhou 350108, China; 2Joint International Research Laboratory on Traffic Psychology & Behaviors, Fuzhou University, Fuzhou 350108, China; 3Department of Civil Engineering, Toronto Metropolitan University, Toronto, ON M5B 2K3, Canada

**Keywords:** safety engineering, tunnel construction, influencing factors, grounded theory, DEMATEL model

## Abstract

Despite the continuous progress of tunnel construction technology and safety management technology, road tunnel construction safety still faces many challenges in China, such as how to ensure the effective management and safety control of people and materials, how to ensure the implementation of technology and program implementation, risk assessment of construction site environmental information, etc. Exploring the causes of tunnel construction accidents and understanding the properties of the factors and their interrelationships can effectively control the sources of risk and contribute to the safety control of tunnel construction. Therefore, we have collected 30 formal accident investigation reports from the government safety supervision and management department from 2005 to 2021, including detailed investigation and accident analysis. Based on grounded theory, a qualitative research method to generalize experience through direct observation, abstraction, and analysis of data, we use Nvivo11 software to analyze reports and obtain 6 selective codes, 16 spindle codes, and 43 open codes. In addition, we construct a theoretical model of tunnel construction accident influencing factors, which passed the saturation test. The Decision-Making Trial and Evaluation Laboratory (DEMATEL) model is used to analyze the influencing mechanism and interaction relationships of these factors. The two dimensions of influence degree and centrality are used to determine the critical influencing factors of tunnel construction accidents in mountainous areas. They are security awareness and professionalism. According to the cause degree, the influencing factors are divided into cause and result factors. Finally, the basis and suggestions for reducing construction accidents are presented.

## 1. Introduction

In recent years, China has paid more and more attention to the strategy of strengthening the country through transportation. As a result, tunnel construction is developing in a blowout way [1], and China has become the country with the most significant number of tunnels. By 2020, the total number of tunnels reached 21,316 [2]. Tunnel construction is a high-risk project with a large scale, high difficulty, long construction period, and difficult construction. Compared with other construction projects, tunnel construction belongs to underground engineering, the construction environment is particularly harsh, often there is only one visible surface, and the rest of the construction is hidden, resulting in hidden, occasional, and difficult to prevent hidden dangers. As the tunnel construction progresses, the surrounding environment also changes homomorphically and is prone to various contingent factors. It has the characteristics of randomness, uncertainty, and many unknown factors [3]. Therefore, despite the continuous development of tunnel construction technology and safety management technology, tunnel construction safety still faces many challenges. The tunnel construction process is very complex, objectively, and involves much machinery and equipment, the construction environment is bad, and affected by natural conditions such as geological, weather, subjectivity of the construction personnel, and professional safety management level, so the tunnel construction site is very prone to accidents, such as landslides, water, construction personnel’s unsafe behavior causing accidents, etc. Such a working environment seriously threatens the life safety of tunnel construction personnel [4,5,6]. According to statistics [7], 109 tunnel construction accidents occurred from 2005–2019, resulting in 338 fatalities. Therefore, identifying and controlling the influencing factors of tunnel construction accidents have become an important issue to be studied.

Many scholars have conducted much research on the risk sources of tunnel accidents. The authors of [8] considered that tunnel failures and collapses could be attributed to design and construction errors. The authors of [9] statistically analyzed 48 cases of tunnel construction accidents from the perspectives of classification, time, regional location, construction methods, and risk sources. The authors of [10] studied three methane explosion accidents from the background, causes, and rescue operations, and put forward prevention measures for methane explosion accidents. A study [11] proposed the concept of tunnel risk assessment and proposed a cost control model based on tunnel risk management, which was applied in many practical projects. The authors of [12] statistically analyzed 97 geological disasters and found that collapse was the main geological disaster in tunnel construction. Another study [13] used the event tree analysis method to analyze the risk of TBM (tunnel boring machine) construction in tunnel construction [14]. The natural, geological, survey design, and construction factors affect the tunnel collapse. The authors of [15] analyzed the construction accident report from two aspects of accident mechanism and accountability and put forward the prevention strategy. These scholars are mainly devoted to studying accident types or a specific type of accident. The complete extraction and analysis of the collision factors of highway tunnel construction accidents are lacking. In [16,17], the authors analyzed the characteristics of tunnel construction accidents in China in terms of temporal distribution, spatial distribution, accident levels and accident types using statistical analysis methods. Authors of another study [18] also counted 10 years of tunnel accident incident cases and used the N-K model to construct a coupled model of tunnel construction risk to reveal the coupling effect between multiple risk factors of tunnel construction accidents.

The research process in the field of construction safety is generally based on the measured performance data of the construction site, and the dynamic system decision-making method, Bayesian network, and other methods are used to dynamically predict and analyze tunnel construction accidents [19,20,21]. However, using qualitative research methods to analyze the relationship between the factors affecting tunnel construction safety are rarely involved, and the guidance to improve tunnel construction safety is limited. Qualitative research is able to analyze the deeper factors hidden behind accidents in terms of behaviors, phenomena, or problems [22]. Therefore, there is a strong need to complement the existing studies on tunnel construction accidents with qualitative studies. In [23,24] both, the triggering factors of unsafe behaviors of tunnel construction workers are based on grounded theory and construct a human factor analysis system for tunnel construction accidents. The concentration of these studies is on human factors, and there is a lack of research exploration on all aspects of tunnel construction.

This paper aims to explore the influencing factors of road tunnel (including urban roads, highways, and expressways) construction accidents in China. The study methodology is shown in Figure 1. First, we collect the official accident investigation reports of government safety production supervision and management departments at all levels in recent years, and the grounded theory model is constructed by extracting, comparing, and summarizing the accident influencing factors using a three-level coding. Then, the Decision-Making Trial and Evaluation Laboratory (DEMATEL) model is used to construct an influence matrix, analyze the importance and causality of the factors, and evaluate the relationships and interactions between them [25]. Finally, the measures and suggestions for reducing construction accidents and improving construction safety are put forward, which have specific theoretical and practical significance.

## 2. Analysis of Safety Factors Using Grounded Theory

### 2.1. Grounded Theory

The grounded theory is an inductive method first proposed by Barney Glaser and Anselm Strauss [26] in 1967 and is widely used in the mechanism study of collision influencing factors [27,28]. The elements of the grounded theory are shown in Figure 2. The research process of grounded theory is to collect relevant raw data and then analyze the data by coders. First, it should be analyzed according to the data order, coded, and refined to form the initial concept. Then, similar concepts are clustered by finding the connection between the initial concepts to determine the main category. Continuing to compare and summarize step by step, we get the core category and the relationship between the categories and finally form the relevant theoretical model [29,30]. The grounded theory model involves three coding processes: open coding, axial coding, and selective coding. Finally, the model is validated using the saturation test.

### 2.2. Data Collection

To prevent and reduce production safety accidents, standardize the reporting and investigation of production safety accidents, and implement the accountability system for production safety accidents, the State Council promulgated the “Regulations on Reporting and Investigation of Production Safety Accidents” [31]. The regulation states that the production of a safety accident report should include an overview of the accident unit; the time, place, and scene of the accident; the brief process of the accident; the number of casualties (including the number of people unaccounted for) and the estimated direct economic losses; measures taken; and so on. Therefore, the tunnel production safety accident report records and analyzes the background, current situation, investigation, and other important contents of the tunnel project in detail, which is the best data for studying the causes of tunnel construction accidents. Therefore, this paper selects the tunnel production safety accident report as the data source of the study. Search and collect accident reports on the official websites of various government departments using keywords such as “tunnel”, “construction”, and “accident report”. Then the accident reports were statistically analyzed, including the geographical location of the accident, the number of casualties, the size of the tunnel, and other factors, and selected representative classic accident reports, including 30 official accident investigation reports from 2011–2021 from government production safety supervision and management departments at all levels. By searching the official websites of various government departments, we collected and screened representative classic accident reports, including 30 official accident investigation reports of production safety supervision and management departments of governments at all levels from 2011 to 2021. The statistical results show that the main types of tunnel construction accidents are collapse, water gushing, mechanical injury, etc., as detailed in Table 1. Table 2 shows the causes of the above tunnel construction accident cases. Referring to the grounded theory saturation principle [32], the ratio of sampling coding samples and theoretical saturation test samples is set to 2:1; that is, 20 coding samples and 10 test samples.

### 2.3. Coding Procedure

#### 2.3.1. Open Coding

The critical technology of grounded theory is “coding data” [33]. Open coding requires researchers to code the original information according to the original appearance with an open mind without any subjective prejudice and theoretical formula. The purpose is to determine the concepts and categories to reflect the social phenomena in the survey [34]. Two coders independently encoded the tunnel construction accident influencing factors as the core and used NVivo11 software to analyze 20 tunnel construction accident case coding sample materials sentence by sentence. The related statements were conceptualized as phrases, and 286 related concepts were obtained. Then, these concepts are repeatedly screened and clustered to obtain 194 initial concepts. According to their correlation, 43 relatively independent open codes, such as safety awareness, professional quality, facilities and equipment, and construction materials, were formed, as shown in Table 3.

#### 2.3.2. Axial Coding

Axial coding is carried out through repeated comparison, analysis, a summary of the content, and concept of the category in selective coding to dig deep into the logical relationship between categories and then develop a more profound organic link to delineate the primary category [35]. The 43 initial concepts in the open coding are clustered and connected through the spindle coding, and 16 main categories are obtained, including safety awareness, professional quality, facilities and equipment, construction materials, engineering hydrogeology, climate and regional environment, engineering investigation, construction drawing design, technical management, quality supervision, safety management, labor management, system construction, monitoring measurement, advanced geological prediction, and dynamic feedback design (see Column 2 of Table 4).

#### 2.3.3. Selective Coding

Selective coding is the last stage of data analysis. The relationship between the core category and the main category is determined [36]. We organize the storylines between the core categories and construct the theoretical framework [37]. In the selective coding stage, the main categories are clustered to obtain the core categories of the influencing factors of tunnel construction accidents. Finally, six-core categories are clustered: human factors, material factors, topographic geology and climate conditions, survey and design, construction management, and information construction (see Column 1 of Table 4).

#### 2.3.4. Model Validation

In order to ensure the validity of the conceptual model, a saturation test is conducted. When new materials do not appear in new conceptual categories, the theoretical model tends to be saturated. The reserved 10 accident reports are used as test materials, and the three-stage coding is carried out according to the same processing steps. The results show that no new concepts are found, they have reached saturation, and the theoretical explanation ability is strong. This indicates that the influencing factors of tunnel construction safety are fully explored and a saturated theoretical model is established, as shown in Table 4.

## 3. Analysis of Safety Factors Using DEMATEL

### 3.1. DEMATEL Model

The DEMATEL model uses graph theory and matrix tools to identify the logical relationship between factors affecting complex systems [38]. This method constructs a direct impact matrix by analyzing the logical connection between the influencing factors in the system. Then the evaluation parameters of each influencing factor are obtained by matrix transformation, including influence degree, affected degree, centrality, and cause degree. to determine the causal relationship between the factors and the position of each factor in the system and identify the critical factors. The DEMATEL model is widely used in the study of the relationship between many elements in a complex system, such as security risk [39], medical management [40], and service evaluation [41]. The steps of the DEMATEL model are as follows:

(1) According to the theoretical model of influencing factors of tunnel construction accidents established based on the grounded theory above, the principal axial coding is selected for questionnaire design, and the order of influencing factors is b_1_, b_2_, …, b*_n_* (*n* = 16).

(2) Invites tunnel experts and scholars to score the strength of the relationship between all factors. The scoring rules [25]: no effect was 0, the weak effect was 1, the moderate effect was 2, and the strong effect was 3. The average score of each expert is taken to obtain the direct influence matrix Z = (b*_ij_*)_*n*×*n*_ between influencing factors.

(3) The normalized direct influence matrix *G* is given by:(1)$$G=\frac{Z}{\max\sum_{i=1}^{n}\sum_{j=1}^{n}b_{ij}}$$

(4) The normalization direct influence matrix *G* is transformed into the comprehensive influence matrix *T* by:(2)$$T=\sum_{k=1}^{n}G^{k}=G{\left(1-G\right)}^{-1}$$

(5) According to the DEMATEL model, the influence degree (*x_i_*), affected degree (*y_i_*), centrality (*m_i_*), and cause degree (*r_i_*) of each factor are calculated as follows:(3)$$\left\{ \begin{array}{l} x_{i}=\sum_{j=1}^{n}\left(t_{ij}\right)\left(1\leq i\leq n\right)\\ y_{i}=\sum_{j=1}^{n}\left(t_{ij}\right)\left(1\leq i\leq n\right)\\ m_{i}=x_{i}+y_{i}\\ r_{i}=x_{i}-y_{i} \end{array}\right.$$
where *t_ij_* represents the combined effect of bi on b*_j_*.

(6) The Cartesian rectangular coordinate system is established with the center as the abscissa, the cause as the ordinate, and o (a, 0) as the origin of the coordinate system, where a is the mean of the center. Then, the critical factors by the coordinate diagram are identified and the relationships between the factors are analyzed.

### 3.2. Construction of Direct Influence Matrix

Based on the results of the grounded theory, we selected 16 factors in axial coding in Table 4 as research objects, numbered them from b_1_ to b_16_, and the numbers corresponded to each other in the order of the table, such as Security awareness to b_1_. We developed an expert questionnaire with this. The invited experts were senior engineers with extensive experience in tunnel construction. We used purposive sampling to select five experts, and then through snowball sampling, looking for more experts, and finally got 20 samples. A total of 20 questionnaires were distributed, and 20 were recovered, with a recovery rate of 100%. Invited experts aged 35–55 (mean = 43.6; SD = 4.6), 10–34 years of employment (mean = 20.1; SD = 6.1), male to female ratio of 7:3.

Through the experts’ evaluation of the influence degree of each influencing factor, the direct influence matrix *Z* was obtained by taking its mean value, and the normalized direct influence matrix *G* was obtained using Equation (2), as shown in Table 5.

### 3.3. Influence Index Calculation

The Excel software transforms the normalized direct influence matrix *G* (Table 4) into a comprehensive influence matrix *T*. According to Equation (3), the influence degree Y, the affected degree X, the centrality M, and the cause degree R of the comprehensive influence matrix are calculated (the specific values and orders are shown in Table 6). Taking the center as the abscissa, the cause as the ordinate, and o (a, 0) (*a* = 9.334) as the intersection of the abscissa and the ordinate, the rectangular coordinate system is established, and the quadrant causality analysis chart is drawn. The critical factors are in the first quadrant.

## 4. Discussion and Recommendations

Through grounded theory and DEMETAL, identifying influencing factors of highway tunnel construction safety is completed. These factors in the pre-construction design, team building, construction safety operations, and other processes require special attention from the relevant parties. Therefore, we make the following analysis of these influencing factors and give appropriate suggestions.

### 4.1. Center Degree Analysis

In the accident system, the importance of each influencing factor b*_i_* is represented by center degree. The greater the center degree, indicating that the greater the impact factor b_i_ is affected by other factors or other factors, the greater the willingness of managers to improve it. Table 5 shows that there are 11 indexes of center degree greater than its geometric average, which are safety management, quality supervision, technical management, advanced geological prediction, monitoring measurement, professional quality, system construction, construction drawing design, safety awareness, and dynamic feedback design. Among them, the centrality of safety management is the largest, indicating that safety management has the highest correlation with other accident-influencing factors. In the accident reports, there are problems of inadequate personnel management at the construction site, such as “At the time of the incident, the personnel of the shield machine in the left line for opening and changing the knife did not match with the personnel listed in the special plan, and the change of personnel was not reported to the supervisory unit for review and approval”. There are mechanical equipment inspection and maintenance safety management is not in place, such as “not mobile and manual power tools use, repair, maintenance inspection, resulting in angle grinder missing protective cover problem was not found in time to solve, does not meet the requirements of use was used for site construction”. At present, safety management is generally considered to be an important factor in controlling risk, and there are also many studies on building safety management systems [42,43] such as the management of equipment [44,45], materials [46,47] and workers [48,49]. In traditional construction site management, it is difficult to organize and coordinate the work of equipment and workers. More advanced information and communication technologies should be introduced to build an efficient management system to improve the efficiency of safety management and guarantee construction safety [50]. At the same time, at the lowest centrality is the regional climate environment, indicating that its interaction with other factors is minimal. The climatic conditions and regional environment of the construction site are not controlled by human activities, so other factors have the least influence on the regional climatic environment, so it has the least central degree.

### 4.2. Influence Degree Analysis

#### 4.2.1. Cause Factor Analysis

Influence degree represents the attribute of influencing factor b*_i_*. According to the positive and negative of cause degree, factors can be divided into cause and result factors. When the influence degree is greater than 0, this factor has a greater impact on other factors and is judged as the cause factor. When the cause degree is less than 0, the factor is more affected by other factors, which is the result factor.

It can be seen from Table 5 that there are five factors in the influencing factor system of tunnel construction safety whose cause degree is greater than 0. According to the order from large to small, they are the regional climate environment, engineering hydrogeology, professional quality, safety awareness, and labor management, which belong to the cause factors, indicating that these factors are easy to affect other factors in the accident system. In the accident report, most of the causes of non-safety accidents are geological environment problems that are difficult to investigate [51], such as “It is inferred that the coupling effect of various unfavorable geological conditions and climatic reasons such as earthquake and sufficient rainfall into April is the main reason for the overall brittle damage and instantaneous and sudden local collapse of the vault.” and “After the accident, the accident investigation team expert group after site investigation, found that the geological conditions are actually striped mixed granite, lamellar development, the existence of kaolin and other unfavorable geological conditions of water softening”. The regional climate environment and engineering hydrogeology are the most significant factors affecting other factors, such as construction, design, and survey. Therefore, the two factors of regional climatic environment and engineering hydrogeology have the highest cause degree. When choosing the tunnel address, one should take into account the influence of geological conditions, while paying attention to the season of construction, in order to prevent the construction project from a lot of rainfall, land freezing, and other adverse effects of the season. At the same time, in controlling tunnel construction safety, it is necessary to focus on cutting off the transmission process between professional quality, safety awareness, labor management, and other factors. According to [52,53], observing the work procedures of construction workers is an effective means of maintaining safe performance in construction projects. The workers’ behavior at work is influenced by their level of professionalism and safety awareness, which is similar to the findings of previous studies in this regard.

#### 4.2.2. Result Factor Analysis

There are 11 factors whose cause degree is less than 0, which belong to the result factors. They are facilities and equipment, monitoring and measurement, advanced geological prediction, technical management, safety management, quality supervision, engineering investigation, dynamic feedback design, system construction, construction materials, and construction drawing design. It shows that these factors are more susceptible to other factors. The construction technology of tunnel projects is usually complicated, so many facilities and construction materials need to be stored in the limited working space. Once the management is improper, it is easy to cause material damage. As in the accident reports, “The direct cause of this accident is: the collapse of the lower section of the formwork due to a combination of defects in the formwork welds, missing flange connection bolts, and excessive speed of concrete pouring”.

The demand for equipment and material management throughout the construction process should be done according to the actual situation on site, and the incoming, use, and deployment of machinery and equipment should be put into place [54]. Monitoring and measurement, advanced geological prediction, engineering investigation, and construction drawing design are all technical operations. This type of survey and design work has a very important impact on the smooth construction of the tunnel, as in the previous study [55]. The completion quality depends on the results of relevant personnel work and is also closely related to the quality of construction work. There is a positive correlation between team competence and skills and construction risk management [56]. Through the management of technology and safety, strict monitoring of construction quality can effectively reduce the risk of tunnel construction accidents. Therefore, when controlling tunnel construction safety, it is necessary to prevent these results from being interfered with by other factors leading to tunnel construction accidents.

### 4.3. Critical Factors Analysis

Figure 3 shows that the critical factors in the first quadrant are safety awareness (b1) and professional quality (b2). As noted, the key to the safety control of tunnel construction is “personnel”, and there are similar conclusions in the relevant tunnel accident studies [57]. In the construction accident reports, “did not organize the relevant personnel of the unit to carry out highway engineering safety supervision and inspection training, resulting in the low quality of safety supervisors, cannot effectively perform their supervisory duties”; “The pump truck operator, with unclear vision and without the command of the pump truck signalman, manually operated the pump truck in violation of the law, which directly led to the accident”, “tunnel construction team safety technical briefing, daily safety education and training system are not implemented, construction personnel on the construction site safety risk awareness is not in place, safety awareness is weak, poor self-safety prevention, is an important cause of the accident”. Therefore, the relevant units of tunnel construction should give more attention to appropriate staff in safety production management. It is necessary to strengthen the cultivation of personnel’s safety awareness and establish a sound safety education system, such as conducting regular safety education activities. Through safety education, the rule consciousness and legal consciousness of construction personnel are cultivated, their attention to safety in production and their sense of responsibility for work are improved, and regulations and laws consciously bind them. At the same time, attention should be paid to developing technical personnel’s professional skills, encouraging sharing of experience between personnel, and improving the technical level and technical standards. Implementing a safety education system can greatly promote tunnel construction safety [2,58].

## 5. Conclusions

Based on the grounded theory, a theoretical model of tunnel construction safety influencing factors with six categories of human factors, material factors, topography, geology and climate conditions, survey and design, construction management, and information construction as the core is constructed. A total of 16 axial codes and 43 open codes are excavated. Combined with the DEMATEL model, the interactions between various factors were analyzed, and the influence degree and center degree of 16 axial codes were calculated. The importance of each influencing factor, the order of influence degree, and its causality were determined. It is a guideline for risk control during tunnel construction and should focus on controlling the most important and influential risk factors. The results show that: (a) safety awareness and professional quality are the critical influencing factors of tunnel construction accidents, where the regional climate environment and engineering hydrogeology are the causes; (b) facilities and equipment, monitoring and measurement, and advanced geological prediction are the result factors.

Based on the research results, relevant suggestions are put forward to take adequate measures to prevent and control the occurrence of accidents in relevant tunnel construction projects. In the control of tunnel construction measures, construction units and other relevant departments should pay special attention to the situation of field personnel, strengthen the education of safety awareness, and the improvement of technical level.

In order to explore the factors influencing tunnel construction accidents as comprehensively as possible, we studied tunnel construction accidents of various sizes of tunnels. However, this study only analyzes tunnel construction accidents that occurred in China, and future analysis of the causes of tunnel construction accidents can be carried out for more regions in order to explore a universal risk control system.

## Figures and Tables

**Figure 1 ijerph-19-16677-f001:**
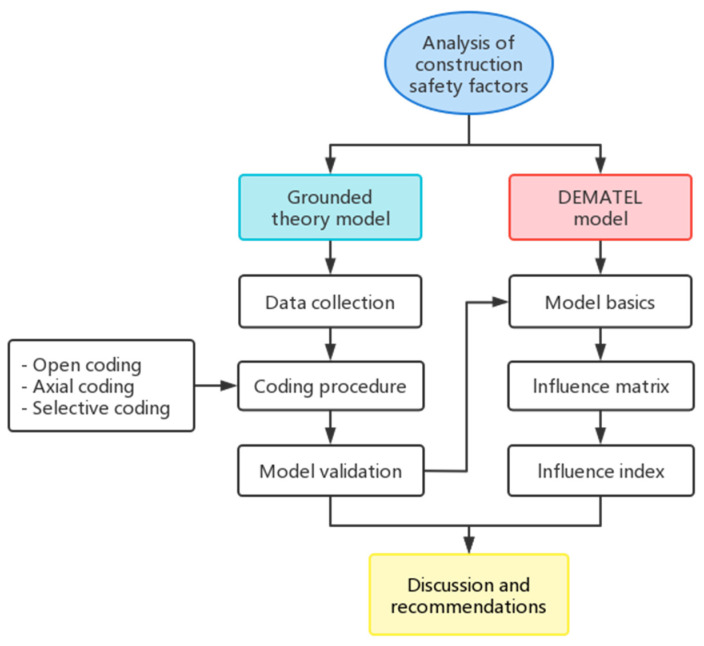
Study methodology.

**Figure 2 ijerph-19-16677-f002:**
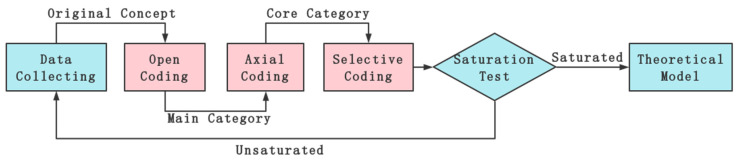
Elements of grounded theory.

**Figure 3 ijerph-19-16677-f003:**
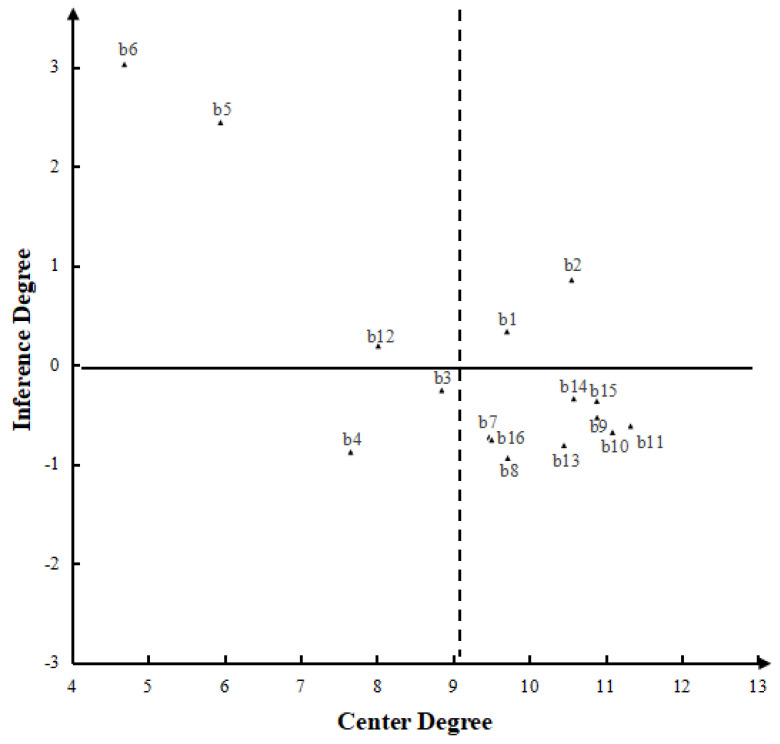
Centrality-causality cartesian coordinate graph.

**Table 1 ijerph-19-16677-t001:** Tunnel construction accident cases.

No.	Date	Tunnel Name	Location (Province)	No. of Injuries	No. of Fatalities	Type	Tunnel Length (m)	Surrounding Rock Type
1	3 May 2014	Longtouling	Anhui	2	6	Collapse	2964	IV
2	6 April 2019	Shantouping	Fujian	0	1	Collapse	3445	Ⅲ
3	29 December 2014	Fenghuangshan	Guangdong	0	5	Collapse	3890	complex
4	24 February 2017	Nanshan Road	Guangdong	0	1	Collapse	167	V
5	22 September 2018	Jiulongling	Fujian	0	1	Mechanical injury	890	complex
6	28 December 2020	Shanggang	Guangxi	0	9	Collapse	2302	complex
7	26 November 2019	Anshi	Yunnan	10	12	Water gushing	5338	complex
8	8 July 2014	Dunliang	Shaanxi	0	3	Collapse	1712	complex
9	30 December 2019	Xichengshan	Shanxi	0	6	Collapse	995	Ⅲ
10	27 August 2019	Hushan	Guangdong	1	3	Collapse	1810	IV
11	16 October 2015	Yanboli	Gansu	0	1	Mechanical injury	3840	complex
12	18 December 2015	Zhoubai	Chongqing	2	6	Collapse	2134	complex
13	18 May 2020	Miraro III	Sichuan	5	6	Water gushing and gas Roof collapse	172	complex
14	5 July 2016	Jinjiayan	Sichuan	0	1	Roof collapse	12,029	V
15	24 February 2015	Wuluo Road I	Sichuan	19	7	Gas explosion	2915	complex
16	5 September 2020	Qipanshi	Fujian	0	1	Roof collapse	2260	Ⅲ
17	16 August 2019	Yongfutun	Guangxi	0	1	Mud inrush	5643.5	V
18	11 September 2021	Puzhuqing I	Yunnan	2	2	Roof collapse	3781	IV
19	5 December 2019	Wangjiazhai	Yunnan	3	3	Mud inrush	8040	V
20	8 October 2021	Pingda	Yunnan	0	1	Collapse	6850	V
21	11 November 2021	Jinao	Zhejiang	1	3	Collapse	2586.7	Ⅱand Ⅲ
22	23 August 2021	Yunwushan II	Chongqing	0	1	Roof collapse	571	V
23	3 November 2019	Yangtaishan	Guangdong	0	1	Collapse	4772	VI
24	10 October 2012	Duoxian	Guangxi	1	5	Collapse	730	complex
25	19 August 2020	Nanjian	Yunnan	1	0	Collapse	2857	V
26	15 July 2020	Shijingshan	Guangdong	0	14	Water gushing	1780	V
27	4 December 2014	Houci	Fujian	21	0	Collapse	1542	V
28	30 August 2021	Jiudianliang	Shaanxi	10	0	Collapse	2195	V and IV
29	22 December 2005	Dongjiashan	Sichuan	11	44	Gas explosion	4089	complex
30	14 July 2018	Baolin	Hubei	6	0	Water gushing	13,840	complex

**Table 2 ijerph-19-16677-t002:** Tunnel construction accident types and causes.

Type	Direct Cause
Collapse	Complex and changeable geological conditions, the tunnel through the fold structure, faults, joint fissure development zone collapse, unstable surrounding rock and lead to loose structure, support is not timely and lead to serious weathering of surrounding rock, failed to make full use of the new Austrian tunneling method to guide the construction, and construction personnel illegal operation.
Water gushing and mud inrush	The surrounding rock of the tunnel is weak and broken, and the surrounding groundwater is enriched. The excavation of the tunnel destroys the pressure balance between the surrounding rock and the groundwater. The stress of the surrounding rock is released in the tunnel direction, and the groundwater moves in the tunnel direction. The surrounding rock is softened and deformed by groundwater erosion, and finally breaks through the critical point to form water and mud inrush.
Roof collapse	Poor geological conditions, construction-induced vibration to increase the concealed closed joints, the construction unit did not change the type of support construction in time, the person in charge did not follow the construction process and requirements, illegal organization of workers risky operations, resulting in the roof rock block off.
Gas explosion	Poor ventilation in the tunnel, gas concentration is too high, encountered high temperature or spark will produce explosion.
Mechanical injury	Construction site dim light, noise, poor environment caused by construction personnel to judge errors, operators with subjective experience to judge the risky operation, mechanical failure.

**Table 3 ijerph-19-16677-t003:** Open coding example (excerpt).

Open Coding	Original Concept
Responsibility	On-site supervision personnel safety consciousness, responsibility is not strong
Technical data management	Technical data management confusion, inspection, and approval data lag, at the same time the construction log content and inspection content do not match
Technical guidance	Smooth blasting lack of professional and technical personnel on-site guidance, there is overbreak phenomenon
Technical tests	The technical disclosure system is not implemented, the disclosure data are incomplete, and there are no technical disclosure data on the safety technical disclosure of primary shotcrete and the two-step excavation method

**Table 4 ijerph-19-16677-t004:** Theoretical model of construction safety factors.

Selective Coding	Axial Coding	Open Coding
Human factors	Security awareness (b_1_)	Responsibility, self-security consciousness, standard operation
	Professional quality (b_2_)	Experienced, skilled, advanced technology
Matter factors	Facilities and equipment (b_3_)	Operation state of new mechanical equipment
	Construction materials (b_4_)	Material quality, material placement
Topographic geology and climatic conditions	Engineering hydrogeology (b_5_)	Particular rock and soil, geological structure, water quantity, and water supply type
	Regional climate environment (b_6_)	Climate conditions, regional environment
Survey and design	Engineering investigation (b_7_)	Survey content and quantity, survey analysis, and report
	Construction drawing design (b_8_)	Design line selection, design depth, design representative
Construction management	Technical management (b_9_)	Technical information management, technical guidance, technical disclosure
	Quality control (b_10_)	Essential process quality inspection, hidden project quality inspection, supervision power allocation
	Safety management (b_11_)	Construction organization design, site safety inspection, and site environment control
	Labor management (b_12_)	Team selection, quality control, and command coordination
	System construction (b_13_)	Education and training system, on-site handover system, risk assessment, and safety management system, emergency response mechanism
Information construction	Monitoring measurement (b_14_)	Monitoring content, quantity, and frequency, monitoring report submission and feedback
	Advanced geological forecast (b_15_)	Methods of advanced geological exploration, submission, and feedback of forecast report
	Dynamic Feedback Design (b_16_)	Dynamic feedback design

b_1_ to b_16_ are the codes of each factor, which will be used later.

**Table 5 ijerph-19-16677-t005:** Direct influence matrix *G* (16 × 16).

*G*	b_1_	b_2_	b_3_	b_4_	b_5_	b_6_	b_7_	b_8_	b_9_	b_10_	b_11_	b_12_	b_13_	b_14_	b_15_	b_16_
b_1_	0.000	1.800	1.400	1.200	0.200	0.100	1.250	1.450	0.241	0.239	0.242	0.145	0.223	0.242	0.236	0.172
b_2_	1.550	0.000	1.350	1.000	0.100	0.100	2.150	2.000	0.297	0.303	0.307	0.183	0.287	0.291	0.286	0.241
b_3_	1.250	1.400	0.000	1.150	0.200	0.300	1.950	1.300	0.178	0.182	0.204	0.072	0.193	0.224	0.240	0.151
b_4_	0.950	0.900	1.150	0.000	0.150	0.200	0.700	1.100	0.089	0.139	0.118	0.058	0.116	0.110	0.098	0.067
b_5_	0.900	0.500	0.950	0.800	0.000	0.700	2.050	2.050	0.158	0.153	0.166	0.065	0.156	0.169	0.193	0.133
b_6_	1.200	0.550	0.850	1.200	2.000	0.000	2.050	1.200	0.125	0.125	0.142	0.070	0.115	0.150	0.140	0.102
b_7_	0.850	1.250	1.400	0.800	1.100	0.250	0.000	2.250	0.182	0.156	0.186	0.071	0.159	0.164	0.180	0.159
b_8_	1.200	1.250	1.150	1.350	0.200	0.150	1.650	0.000	0.211	0.212	0.225	0.087	0.201	0.205	0.208	0.189
b_9_	1.400	1.850	1.600	1.100	0.300	0.250	1.950	1.850	0.184	0.243	0.247	0.119	0.236	0.239	0.242	0.202
b_10_	1.700	1.900	1.700	2.050	0.100	0.150	1.550	1.800	0.235	0.173	0.251	0.159	0.253	0.210	0.211	0.149
b_11_	2.100	1.600	1.850	1.650	0.200	0.350	1.700	1.700	0.241	0.262	0.183	0.154	0.241	0.244	0.243	0.158
b_12_	2.150	1.600	1.050	1.250	0.300	0.250	0.900	0.750	0.126	0.156	0.147	0.068	0.139	0.112	0.112	0.085
b_13_	1.500	1.700	1.250	1.650	0.400	0.200	1.050	1.250	0.210	0.246	0.211	0.108	0.156	0.207	0.208	0.168
b_14_	1.400	1.700	1.400	1.050	1.200	0.200	1.500	1.700	0.218	0.208	0.223	0.089	0.221	0.155	0.223	0.183
b_15_	1.600	1.550	1.600	1.200	1.750	0.300	1.800	1.800	0.225	0.208	0.225	0.086	0.218	0.220	0.166	0.214
b_16_	1.100	1.250	1.200	1.400	0.400	0.350	1.500	1.900	0.207	0.184	0.187	0.078	0.193	0.189	0.195	0.111

**Table 6 ijerph-19-16677-t006:** Centrality and cause degree.

Influenced Degree Y		Influence Degree X		Center Degree M ^a^		Cause Degree R	
Order	Value	Order	Value	Order	Value	Order	Value
b_11_	5.970	b_11_	11.328	b_11_	11.328	b_6_	3.030
b_10_	5.884	b_10_	11.091	b_10_	11.091	b_5_	2.445
b_9_	5.707	b_9_	10.888	b_9_	10.888	b_2_	0.858
b_13_	5.631	b_15_	10.885	b_15_	10.885	b_1_	0.339
b_15_	5.624	b_14_	10.582	b_14_	10.582	b_12_	0.195
b_14_	5.460	b_2_	10.554	b_2_	10.554	b_3_	−0.253
b_8_	5.326	b_13_	10.454	b_13_	10.454	b_14_	−0.337
b_16_	5.129	b_8_	9.716	b_8_	9.716	b_15_	−0.362
b_7_	5.099	b_1_	9.706	b_1_	9.706	b_9_	−0.526
b_2_	4.848	b_16_	9.505	b_16_	9.505	b_11_	−0.613
b_1_	4.683	b_7_	9.472	b_7_	9.472	b_10_	−0.676
b_3_	4.551	b_3_	8.850	b_3_	8.850	b_7_	−0.726
b_4_	4.265	b_12_	8.017	b_12_	8.017	b_16_	−0.753
b_12_	3.911	b_4_	7.655	b_4_	7.655	b_13_	−0.808
b_5_	1.752	b_5_	5.948	b_5_	5.948	b_4_	−0.875
b_6_	0.828	b_6_	4.686	b_6_	4.686	b_8_	−0.936

^a^ The geometric mean of the Center Degree M is 9.334.

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
