# Peer review of "Factors Affecting Road Tunnel Construction Accidents in China Based on Grounded Theory and DEMATEL"

_ijerph, 2022, doi:10.3390/ijerph192416677_

Round 1

Reviewer 1 Report

Review report. Paper: “Factors Affecting Tunnel Construction Accident Based on Grounded Theory and DEMATEL” (Yang et al.)

This article focuses on the analysis of 30 accident reports related to tunnel construction in China. The analysis is done first by analyzing the content of the reports in order to create a conceptual model including 6 core categories and 16 selective categories. Based on the selective categories, the DEMATEL method is used to to analyze the influencing mechanism and interaction relationships of these factors. Results suggests that “safety awareness and professional quality are the critical influencing factors of tunnel construction accidents, where the regional climate environment and engineering hydrogeology are the causes. Facilities and equipment, monitoring and measurement, advanced geological prediction are the result factors”.

The initial objective of the article, namely a more qualitative analysis of accidents related to tunnel construction, is relevant. This is a particularly risky activity. The methodological aspect is particularly put forward, even if there are some shortcomings in particular on the constitution of the sample of investigation reports (see comments). The methodological aspect takes up a lot of space to the detriment of the discussion. The discussions remain at a high theoretical level which somewhat diminishes the impact of the paper. It would have been interesting to come back to a qualitative level with, in particular, concrete examples to support and discuss the results.

Abstract:

-        Line 12 : What are the “challenges”? You can be more specific (I have several comments like this)

-        Line 13 : Specify what types of accidents, in which geographical region.

-        The objective is not clearly stated in the summary.

-        Line 15: Explain in few words what is « grounded theory ».

-        Line 16: The theoretical model is made up of the different categories. The wording of the sentence suggests that it is an extra step. to be rephrased.

1.      Introduction

-        What is considered a tunnel construction site in this study (road, rail, metro, others, all). What are the inclusion and exclusion criteria? This study is specific to china you should mention it.

-        Are there any relevant differences to mention between the construction of tunnels in China and elsewhere?

-        Be more specific about the risks inherent in these sites and what distinguishes them from others. Ex. Line 33: “It has the characteristics of randomness, uncertainty and many unknown factors”. Be more specific, this award can apply to all major construction sites.

-        Line 43 and following: What are the risks considered in the study? What is considered a tunnel accident?

-        Line 68: What is the originality of the study compared to other studies? The qualitative aspect is mentioned just before. What is the benefit of such an approach in the context?

-        Lines 69 -77: Refer to sections where more details are given. This helps guide the reader.

-        Figure 1: Reading the rest of the article, it seems to me that the DEMATEL model follows on from the grounded theory model by using selective coding.

2. Analysis of Safety Factors Using Grounded Theory

-        End of section 2.1: mention that details on each of the steps are following.

-        Section 2.2: There is a lack of details on the selection of the 30 reports. This is the raw data of the study. It’s important to give more information in particular to judge the generalizability of the study

-        How was the sample drawn? How representative is it? What are the inclusion/exclusion criteria? What is investigated and what is not? Fatal accident only? If not why an incident is being investigated?

-        Table 2: Are the causes stated in the report or is it an extrapolation of team members?

-        Lines 115-118: These explanations are not clear without more details. Put this information in section 2.4.4 instead.

-        Table 2: What is the “Austrian principle”?

-        Table 4: Change column titles for: core categories (selective coding), main category (axial coding), open code (open coding). Try to harmonize the terms used throughout the article (eg. Abstract: spindle).

3. DEMATEL

-        How was the sample of experts defined? By which method (e.g. judgmental sampling)? What were the inclusion/exclusion criteria? Is 20 experts enough and why? The rest of the results depend on the credibility of the experts.

-        I cannot comment on the DEMATEL method, it is outside my field of competence. Table 5: Shouldn't a normalized matrix have values less than 1? Is it not rather the Z matrix that is presented? (sorry if I'm wrong).

-        Tables 5, 6 and Figure 3: It would have been relevant to list b1-b16 in table 4 in order to better understand tables 5 and 6 as well as figure 3.

4. Discussions

See my comment from the beginning: “The discussions remain at a high theoretical level which somewhat diminishes the impact of the paper. It would have been interesting to come back to a qualitative level with, in particular, concrete examples to support and discuss the results.” It's not very precise as a comment, but that was my feeling when reading the discussion.

Misprints, other remarks :

-        Line 15 : Delete “and the code”

-        Line 56: Delete one “)”

-        Lines 149-15: There no verb in this sentence

-        Tables 3 and 4: The fact that the text is centered vertically, in the absence of horizontal dividing lines, makes it difficult to read.

-        171-173: Revise the sentence. Split “and critical factors” into another sentence?

Author Response

Response to Reviewer 1 Comments

Point 1: What are the "challenges"? You can be more specific (I have several comments like this)

Response 1: Thank you for your valuable comments. We explain the challenges as follows(Lines 12-14).

"such as how to ensure the effective management and safety control of people and materials, how to ensure the implementation of technology and program implementation, risk assessment of construction site environmental information, etc."

Point 2: Specify what types of accidents, in which geographical region.

Response 2: Thank you for your valuable comments. We did not specify a particular type of accident. Tunnels are distributed in various provinces of China, mainly in hilly and mountainous areas. Our tunnel construction accident is defined as the tunnel construction process, the occurrence of casualties or property losses, such as accidents caused by collapse, accidents caused by improper operation of workers.

Point 3: The objective is not clearly stated in the summary.

Response 3: Thank you for your valuable comments. We have added the objective as follows(Lines 15-17):

"Exploring the causes of tunnel construction accidents and understanding the properties of the factors and their interrelationships can effectively control the sources of risk and contribute to the safety control of tunnel construction."

Point 4:  Explain in few words what is « grounded theory ».

Response 4: Grounded theory is simply as follows (Lines 19-21):

"a qualitative research method to generalize experience through direct observation, abstraction, and analysis of data."

Details in 2.1.Grounded Theory (Lines 133-143): 

"The grounded theory is an inductive method first proposed by Barney Glaser and Anselm Strauss [26] in 1967 and is widely used in the mechanism study of collision influ-encing factors [27,28]. The elements of the grounded theory are shown in Figure 2. The re-search process of grounded theory is to collect relevant raw data and then analyze the da-ta by coders. First, it should be analyzed according to the data order, coded and refined to form the initial concept. Then, the similar concepts are clustered by finding the connection between the initial concepts to determine the main category. Continuing to compare and summarize step by step, we get the core category and the relationship between the catego-ries, and finally form the relevant theoretical model [29,30]. The grounded theory model involves three coding processes: open coding, axial coding, and selective coding. Finally, the model is validated using the saturation test."

Point 5: The theoretical model is made up of the different categories. The wording of the sentence suggests that it is an extra step. to be rephrased.

Response 5: In the use of grounded theory research, ' theoretical model ' is generally used to describe the results, the theoretical model is appropriate, such as:

Moran M, Baron-Epel O, Assi N. Causes of road accidents as perceived by Arabs in Israel: A qualitative study. Transportation research part F: traffic psychology and behaviour, 2010, 13(6): 377-387.

Point 6: What is considered a tunnel construction site in this study (road, rail, metro, others, all). What are the inclusion and exclusion criteria? This study is specific to china you should mention it.

Response 6: Thank you for your valuable comments. This study focuses on the construction accident cases of highway tunnels in China, including tunnels in urban roads, highways and expressways. The reason for choosing the road tunnel construction accident is that the section of the highway tunnel is relatively larger than that of the railway tunnel, and the number and construction difficulty are greater than the railway tunnel.  Therefore, the choice of road tunnel is more representative.

We have added descriptions of highway tunnels in China as follows (Lines 98-99):

"This paper aims to explore the influencing factors of road tunnel (including urban roads, highways and expressways) construction accidents in China."

Point 7: Are there any relevant differences to mention between the construction of tunnels in China and elsewhere?

Response 7: Thank you for your valuable comments. There are differences in design concept, design specification and construction technology between China and other countries in tunnel construction. This paper only studies tunnel construction accidents in China, which may not be targeted in other regions. This is also our future research direction.

Point 8: Be more specific about the risks inherent in these sites and what distinguishes them from others. Ex. Line 33: "It has the characteristics of randomness, uncertainty and many unknown factors". Be more specific, this award can apply to all major construction sites.

Response 8: Thank you for your valuable comments. We added a comparison of the tunnel construction site with other construction sites as follows (Lines 39-45) :

"Compared with other construction projects, tunnel construction belongs to underground engineering, the construction environment is particularly harsh, often there is only one visible surface, the rest of the construction is hidden, resulting in hidden, occasional and difficult to prevent hidden dangers. As the tunnel construction progresses, the surrounding environment also changes homomorphically and is prone to various contingent factors."

Point 9:  Line 43 and following: What are the risks considered in the study? What is considered a tunnel accident?

Response 9: Thank you for your valuable comments. We define the tunnel construction accident is in the tunnel construction process, the occurrence of casualties or property losses, such as collapse caused by accidents, accidents caused by improper operation of workers. According to the results of grounded theory, the risks are explored mainly from six aspects : human factors, factorster, Topographic geology and climatic conditions, Survey and design management, Construction management, and Information construction. Details in Table 4.(Lines 257).

Point 10: Line 68: What is the originality of the study compared to other studies? The qualitative aspect is mentioned just before. What is the benefit of such an approach in the context?

Response 10: Thank you for your valuable comments. We collected the tunnel accident report and used it as the research data to explore the causes of tunnel construction accidents by using the grounded theory method, and distinguished the existence and strength of the causal relationship between the factors through the dematel model. Through the calculation of the two core indicators of centrality and cause, the causal correlation between the factors is revealed, and the importance of each influencing factor in the system is ranked. Qualitative research can analyze the deep factors hidden behind it from the perspective of behavior, phenomenon or problem, which is more suitable for the theme of this study. The grounded theory absorbs the advantages of quantitative research in qualitative research. With rigorous and systematic research procedures, it uses deductive induction to solve the problems of lack of generalization, replication, accuracy, rigor and verifiability in qualitative research, and realizes the 'scientific nature' of research in qualitative research.

We have added the following (Lines 90-93):

"Qualitative research is able to analyze the deeper factors hidden behind tunnel construction accidents in terms of behaviors, phenomena or problems[22]. Therefore, there is a strong need to complement the existing studies on tunnel construction accidents with qualitative studies."

  1. Mohajan, H.K. Qualitative research methodology in social sciences and related subjects. Journal of economic development, environment and people 2018, 7, 23-48.

Point 11: Lines 69 -77: Refer to sections where more details are given. This helps guide the reader.

Response 11: Thank you for your valuable comments. We added details as follows (Lines 103-106):

"the Decision-Making Trial and Evaluation Laboratory (DEMATEL) model is used to construct an influence matrix, analyze the importance and causality of the factors, and evaluate the relationships and interactions between them."

Point 12: Figure 1: Reading the rest of the article, it seems to me that the DEMATEL model follows on from the grounded theory model by using selective coding.

Response 12: Thank you for your valuable comments. The selection of indicators for the Dematel model is based on the results of grounded theory research.

Point 13: Section 2.2: There is a lack of details on the selection of the 30 reports. This is the raw data of the study. It's important to give more information in particular to judge the generalizability of the study. How was the sample drawn? How representative is it? What are the inclusion/exclusion criteria? What is investigated and what is not? Fatal accident only? If not why an incident is being investigated?

Response 13: Thank you for your valuable comments. We collected tunnel construction accident reports from the official websites of various government departments, and then conducted statistical analysis of information on the geographical location of the accidents, the number of casualties, and the size of the tunnels, and selected cases of tunnel construction accidents of various construction scales for study. Screened incident reports including 30 official accident investigation reports of production safety supervision and management departments of governments at all levels from 2011 to 2021. The statistical results show that the main types of tunnel construction accidents are collapse, water gushing, mechanical injury, etc., as detailed in Table 1(Line 163).

Point 14:  Table 2: Are the causes stated in the report or is it an extrapolation of team members?

Response 14: The immediate cause in table 2 is that members reviewed all accident reports, and the coders excerpted statements from the report.

Point 15: Lines 115-118: These explanations are not clear without more details. Put this information in section 2.4.4 instead.

Response 15: Thank you for your valuable comments. Lines 115-118 present 'Regulations on Reporting and Investigation of Production Safety Accidents', which we explain later, and why we conducted our research with accident reports, as follows(Lines 170-177):

"The regulation provides that production safety accident report should include an overview of the accident unit; the time, place and scene of the accident; the brief process of the accident; the number of casualties (including the number of people unaccounted for) and the estimated direct economic losses; measures taken and so on. Therefore, the tunnel production safety accident report records and analyzes the background, current situation, investigation and other important contents of the tunnel project in detail, which is the best data for studying the causes of tunnel construction accidents."

In the Model Validation section, the research data source is also these accident reports, so it is not explained again.

Point 16: Table 2: What is the "Austrian principle"?

Response 16: Thank you for your valuable comments. The 'Austrian principle' should be ' new austrian tunnelling method ', a tunnel construction method. We corrected it in the paper.

Point 17:  Table 4: Change column titles for: core categories (selective coding), main category (axial coding), open code (open coding). Try to harmonize the terms used throughout the article (eg. Abstract: spindle).

Response 17: Thank you for your valuable comments. We re-examined the article and unified the terminology. In order to make the table correspond to the previous content, the title table 4 of the corresponding section is used as the column title.

Point 18: How was the sample of experts defined? By which method (e.g. judgmental sampling)? What were the inclusion/exclusion criteria? Is 20 experts enough and why? The rest of the results depend on the credibility of the experts.

Response 18: Thank you for your valuable comments. The invited experts are senior engineers with extensive experience in tunnel construction. We used the purpose sampling to select five experts, and then through the snowball sampling, looking for more experts, and finally got 20 samples. The DEMATEL method has low requirements for the number of samples, and good research results can be obtained using small samples. 20 samples can meet the research needs.

Lin Y-T, Yang Y-H, Kang J-S, Yu H-C. Using DEMATEL method to explore the corecompetences and causal effect of the IC design service company: An empirical casestudy. Expert Systems with Applications, 2011, 38(5): 6262-6268.

Hsu C-W, Kuo T-C, Chen S-H, Hu A H. Using DEMATEL to develop a carbon management model of supplier selection in green supply chain management. Journal of Cleaner Production, 2013, 56(0): 164-172.

Point 19:  I cannot comment on the DEMATEL method, it is outside my field of competence. Table 5: Shouldn't a normalized matrix have values less than 1? Is it not rather the Z matrix that is presented? (sorry if I'm wrong).

Response 19: Z direct impact matrix is the average score of each expert, G is the normalized matrix. Because of the limited space, and they reflect the same regulation, we only selected the Z matrix in the paper.

Point 20: Tables 5, 6 and Figure 3: It would have been relevant to list b1-b16 in table 4 in order to better understand tables 5 and 6 as well as figure 3.

Response 20: Thank you for your valuable comments. We have added explanations as follows (Lines 332-335):

"Based on the results of the grounded theory, we selected 16 factors in axial coding in Table 4 as research objects, numbered them b1 to b16, and the numbers corresponded to each other in the order of the table, such as Security awareness as b1. And developed an expert questionnaire with this. "

Point 21: See my comment from the beginning: "The discussions remain at a high theoretical level which somewhat diminishes the impact of the paper. It would have been interesting to come back to a qualitative level with, in particular, concrete examples to support and discuss the results." It's not very precise as a comment, but that was my feeling when reading the discussion.

Response 21: Thank you for your valuable comments. We have refined the discussion section as follows (Lines 384-403, 419-429, 439-448):

"At present, safety management is generally considered to be an important factor in controlling risk, and there are also many studies on building safety management systems [42,43], Such as the management of equipment [44,45], materials [46,47] and workers [48,49]. In the traditional construction site management, it is difficult to organize and coordinate the work of equipment and workers. More advanced information and communication technologies should be introduced to build an efficient management system to improve the efficiency of safety management and guarantee construction safety [50]. At the same time, the lowest centrality is the regional climate environment, indicating that its interaction with other factors is minimal. The climatic conditions and regional environment of the construction site are not controlled by human activities, so other factors have the least influence on the regional climatic environment, so it has the least center degree."

"Therefore, the two factors of regional climatic environment and engineering hydrogeology have the highest cause degree. When choosing the tunnel address should take more into account the influence of geological conditions, while paying attention to the season of construction, in order to prevent the construction project from a lot of rainfall, land freezing and other adverse effects by the season. At the same time, in controlling tunnel construction safety, it is necessary to focus on cutting off the transmission process between professional quality, safety awareness, labor management, and other factors. According to [52] and [53], observing the work procedures of construction workers is an effective means of maintaining safe performance in construction projects. The workers' behavior at work is influenced by their level of professionalism and safety awareness, which is similar to the findings of previous studies in this regard."

"The demand for equipment and material management throughout the construction process should be done according to the actual situation on site, and the incoming, use and deployment of machinery and equipment should be put into place [54] Monitoring and measurement, advanced geological prediction, engineering investigation and construction drawing design are all technical operations. This type of survey and design work has a very important impact on the smooth construction of the tunnel, as the previous study [55].The completion quality depends on the results of relevant personnel work, and is also closely related to the quality of construction work. Positive correlation between team competence and skills and construction risk management [56]."

  1. Teizer, J.; Allread, B.S.; Fullerton, C.E.; Hinze, J. Autonomous pro-active real-time construction worker and equipment operator proximity safety alert system. Automation in construction 2010, 19, 630-640.
  2. Yang, H.; Chew, D.A.; Wu, W.; Zhou, Z.; Li, Q. Design and implementation of an identification system in construction site safety for proactive accident prevention. Accident Analysis & Prevention 2012, 48, 193-203.
  3. Yang, J.; Vela, P.; Teizer, J.; Shi, Z. Vision-based tower crane tracking for understanding construction activity. Journal of Computing in Civil Engineering 2014, 28, 103-112.
  4. Chae, S.; Kano, N. A location system with RFID technology in building construction site. In Proceedings of the Proceedings of the 22nd International Symposium on Automation and Robotics in Construction, 2005; pp. 1-6.
  5. Grau, D.; Caldas, C.H.; Haas, C.T.; Goodrum, P.M.; Gong, J. Assessing the impact of materials tracking technologies on construction craft productivity. Automation in construction 2009, 18, 903-911.
  6. Caldas, C.H.; Torrent, D.G.; Haas, C.T. Using global positioning system to improve materials-locating processes on industrial projects. Journal of Construction Engineering and Management 2006, 132, 741-749.
  7. Carbonari, A.; Giretti, A.; Naticchia, B. A proactive system for real-time safety management in construction sites. Automation in construction 2011, 20, 686-698.
  8. Teizer, J.; Cheng, T.; Fang, Y. Location tracking and data visualization technology to advance construction ironworkers' education and training in safety and productivity. Automation in Construction 2013, 35, 53-68.
  9. Zhou, H.; Wang, H.; Zeng, W. Smart construction site in mega construction projects: A case study on island tunneling project of Hong Kong-Zhuhai-Macao Bridge. Frontiers of Engineering Management 2018, 5, 78-87.
  10. Abdelhamid, T.S.; Everett, J.G. Identifying root causes of construction accidents. Journal of construction engineering and management 2000, 126, 52-60.
  11. Li, H.; Chan, G.; Huang, T.; Skitmore, M.; Tao, T.Y.; Luo, E.; Chung, J.; Chan, X.; Li, Y. Chirp-spread-spectrum-based real time location system for construction safety management: A case study. Automation in construction 2015, 55, 58-65.
  12. Samee, K.; Pongpeng, J. Structural equation model for construction equipment management affecting project and corporate performance. KSCE Journal of Civil Engineering 2016, 20, 1642-1656.
  13. Xiang, Y.; Yang, Y. Challenge in design and construction of submerged floating tunnel and state-of-art. Procedia engineering 2016, 166, 53-60.
  14. Moe, T.L.; Pathranarakul, P. An integrated approach to natural disaster management: public project management and its critical success factors. Disaster Prevention and Management: An International Journal 2006, 15, 396-413.

Point 22:  Misprints, other remarks :

-        Line 15 : Delete "and the code"

-        Line 56: Delete one ")"

-        Lines 149-15: There no verb in this sentence

-        Tables 3 and 4: The fact that the text is centered vertically, in the absence of horizontal dividing lines, makes it difficult to read.

-        171-173: Revise the sentence. Split "and critical factors" into another sentence?

Response 22: Thank you for your valuable comments. We're sorry for these mistakes, we once again carefully reviewed the details of the article, corrected similar mistakes.

Reviewer 2 Report

1. Abstract: Rather than being informative, general and intangible expressions are available.

2. Introduction: Authors say that "However, using qualitative research methods to analyze the relationship between the factors affecting tunnel construction safety is rarely involved, and the guidance to improve tunnel construction safety is limited." Words such as rarely and limited indicate the availability of similar studies in the literature. What is the difference between this study and these researches?

3. Both grounded theory and dematel are well-known methods that have been used for a very long time in academia and even in the undergraduate level. If so, what is the methodological contribution of this study to the related literature?

4. Chapter 4 is very infertile. It neither compares its findings and results to those of similar previous studies nor includes a large discussion.

5. Conclusions: Includes very general expressions instead of giving specific results and concluding remarks. Moreover, it does not contain limitations and implications such as research, practical, and social.

6. References: In the literature, there are very similar papers but many of them are not included in this study despite the fact that some of them have been performed in China. Just a few of them are as follows:

Pan et al. (2022) Coupling Analysis to Investigate Multiple Risk Factors for Tunnel Construction Accidents Based on N-K Model. Tunnel Construction, 42(9), pp. 1537-154.

Zhu et al. (2022) Statistical analysis of major tunnel construction accidents in China from 2010 to 2020. Tunnelling and Underground Space Technology, 124, 104460.

Xin et al. (2019) Construction of accident rate model for tunnel group sections of expressway in mountainous areas. IOP Conference Series: Materials Science and Engineering, 688(4), 044009.

Zhang et al. (2018) Law and Characteristics Analysis of Domestic Tunnel Construction Accidents from 2006 to 2016. Modern Tunnelling Technology, 55(3), pp. 10-17.

Author Response

Response to Reviewer 2 Comments

Point 1: Abstract: Rather than being informative, general and intangible expressions are available.

Response 1: Thank you for your valuable comments. We modified the abstract as follows (Lines 11-29):

“Despite the continuous progress of tunnel construction technology and safety management tech-nology, tunnel construction safety still faces many challenges, such as how to ensure the effec-tive management and safety control of people and materials, how to ensure the implementation of technology and program implementation, risk assessment of construction site environmental information, etc. Exploring the causes of tunnel construction accidents and understanding the properties of the factors and their interrelationships can effectively control the sources of risk and contribute to the safety control of tunnel construction. Therefore, we have collected 30 for-mal accident investigation reports from the government safety supervision and management department from 2005 to 2021, including detailed investigation and accident analysis. Based on grounded theory, a qualitative research method to generalize experience through direct obser-vation, abstraction, and analysis of data, we use Nvivo11 software to analyze reports and obtain 6 selective codes, 16 spindle codes, and 43 open codes. In addition, we construct a theoretical model of tunnel construction accident influencing factors, which passed the saturation test. The Decision-Making Trial and Evaluation Laboratory (DEMATEL) model is used to analyze the in-fluencing mechanism and interaction relationships of these factors. The two dimensions of influ-ence degree and centrality are used to determine the critical influencing factors of tunnel con-struction accidents in mountainous areas. They are security awareness and professionalism. Ac-cording to the cause degree, the influencing factors are divided into cause and result factors. Fi-nally, the basis and suggestions for reducing construction accidents are presented.”

Point 2: Introduction: Authors say that "However, using qualitative research methods to analyze the relationship between the factors affecting tunnel construction safety is rarely involved, and the guidance to improve tunnel construction safety is limited." Words such as rarely and limited indicate the availability of similar studies in the literature. What is the difference between this study and these researches?

Response 2: Thank you for your valuable comments. The existing research on the influencing factors of tunnel construction accidents based on qualitative research mainly focuses on human factors. Our research considers more dimensions and more comprehensive in the tunnel construction process.We have added relevant content in the paper, as follows(Lines 93-97):

“[23] and [24] both study the triggering factors of unsafe behaviors of tunnel construction workers based on grounded theory and construct a human factor analysis system for tunnel construction accidents. The concentration of these studies is on human factors, and there is a lack of research exploration on all aspects of tunnel construction.”

  1. Yun, C.; Yu-xin, L. Research on the Causes of Traffic Tunnel ConstructionSafety Accidents Based on HFACS. Journal of Engineering Management 2021, 67-72.
  2. Junwen, M.; Ruirui, W. Research on early warning of unsafe behavior of high-altitude tunnel builders based on HLM-BP. Journal of Railway Science and Engineering 2022, 1-12.

Point 3: Both grounded theory and dematel are well-known methods that have been used for a very long time in academia and even in the undergraduate level. If so, what is the methodological contribution of this study to the related literature?

Response 3: Thank you for your valuable comments. Grounded theory and dematel is mainly used in the field of humanities and economic management. We apply it to the field of tunnel engineering and enrich the application field of the method.

Point 4: Chapter 4 is very infertile. It neither compares its findings and results to those of similar previous studies nor includes a large discussion.

Response 4: We have refined the discussion section as follows(Lines 384-403, 419-429, 439-448):

“At present, safety management is generally considered to be an important factor in controlling risk, and there are also many studies on building safety management systems [42,43], Such as the management of equipment [44,45], materials [46,47] and workers [48,49]. In the traditional construction site management, it is difficult to organize and coordinate the work of equipment and workers. More advanced information and communication technologies should be introduced to build an efficient management system to improve the efficiency of safety management and guarantee construction safety [50]. At the same time, the lowest centrality is the regional climate environment, indicating that its interaction with other factors is minimal. The climatic conditions and regional environment of the construction site are not controlled by human activities, so other factors have the least influence on the regional climatic environment, so it has the least center degree.”

“Therefore, the two factors of regional climatic environment and engineering hydrogeology have the highest cause degree. When choosing the tunnel address should take more into account the influence of geological conditions, while paying attention to the season of construction, in order to prevent the construction project from a lot of rainfall, land freezing and other adverse effects by the season. At the same time, in controlling tunnel construction safety, it is necessary to focus on cutting off the transmission process between professional quality, safety awareness, labor management, and other factors. According to [52] and [53], observing the work procedures of construction workers is an effective means of maintaining safe performance in construction projects. The workers' behavior at work is influenced by their level of professionalism and safety awareness, which is similar to the findings of previous studies in this regard.”

“The demand for equipment and material management throughout the construction process should be done according to the actual situation on site, and the incoming, use and deployment of machinery and equipment should be put into place [54] Monitoring and measurement, advanced geological prediction, engineering investigation and construction drawing design are all technical operations. This type of survey and design work has a very important impact on the smooth construction of the tunnel, as the previous study [55].The completion quality depends on the results of relevant personnel work, and is also closely related to the quality of construction work. Positive correlation between team competence and skills and construction risk management [56].”

  1. Teizer, J.; Allread, B.S.; Fullerton, C.E.; Hinze, J. Autonomous pro-active real-time construction worker and equipment operator proximity safety alert system. Automation in construction 2010, 19, 630-640.
  2. Yang, H.; Chew, D.A.; Wu, W.; Zhou, Z.; Li, Q. Design and implementation of an identification system in construction site safety for proactive accident prevention. Accident Analysis & Prevention 2012, 48, 193-203.
  3. Yang, J.; Vela, P.; Teizer, J.; Shi, Z. Vision-based tower crane tracking for understanding construction activity. Journal of Computing in Civil Engineering 2014, 28, 103-112.
  4. Chae, S.; Kano, N. A location system with RFID technology in building construction site. In Proceedings of the Proceedings of the 22nd International Symposium on Automation and Robotics in Construction, 2005; pp. 1-6.
  5. Grau, D.; Caldas, C.H.; Haas, C.T.; Goodrum, P.M.; Gong, J. Assessing the impact of materials tracking technologies on construction craft productivity. Automation in construction 2009, 18, 903-911.
  6. Caldas, C.H.; Torrent, D.G.; Haas, C.T. Using global positioning system to improve materials-locating processes on industrial projects. Journal of Construction Engineering and Management 2006, 132, 741-749.
  7. Carbonari, A.; Giretti, A.; Naticchia, B. A proactive system for real-time safety management in construction sites. Automation in construction 2011, 20, 686-698.
  8. Teizer, J.; Cheng, T.; Fang, Y. Location tracking and data visualization technology to advance construction ironworkers' education and training in safety and productivity. Automation in Construction 2013, 35, 53-68.
  9. Zhou, H.; Wang, H.; Zeng, W. Smart construction site in mega construction projects: A case study on island tunneling project of Hong Kong-Zhuhai-Macao Bridge. Frontiers of Engineering Management 2018, 5, 78-87.
  10. Abdelhamid, T.S.; Everett, J.G. Identifying root causes of construction accidents. Journal of construction engineering and management 2000, 126, 52-60.
  11. Li, H.; Chan, G.; Huang, T.; Skitmore, M.; Tao, T.Y.; Luo, E.; Chung, J.; Chan, X.; Li, Y. Chirp-spread-spectrum-based real time location system for construction safety management: A case study. Automation in construction 2015, 55, 58-65.
  12. Samee, K.; Pongpeng, J. Structural equation model for construction equipment management affecting project and corporate performance. KSCE Journal of Civil Engineering 2016, 20, 1642-1656.
  13. Xiang, Y.; Yang, Y. Challenge in design and construction of submerged floating tunnel and state-of-art. Procedia engineering 2016, 166, 53-60.
  14. Moe, T.L.; Pathranarakul, P. An integrated approach to natural disaster management: public project management and its critical success factors. Disaster Prevention and Management: An International Journal 2006, 15, 396-413.

Point 5: Conclusions: Includes very general expressions instead of giving specific results and concluding remarks. Moreover, it does not contain limitations and implications such as research, practical, and social.

Response 5: Thank you for your valuable comments. We modify the conclusion as follows (Lines 480-481, 491-495):

“It is a guideline for risk control during tunnel construction and should focus on controlling the most important and influential risk factors.”

“In order to explore the factors influencing tunnel construction accidents as compre-hensively as possible, we studied tunnel construction accidents in various sizes tunnels. However, this study only analyzes tunnel construction accidents that occurred in China, and future analysis of the causes of tunnel construction accidents can be carried out for more regions in order to explore a universal risk control system.”

Point 6: 6.References: In the literature, there are very similar papers but many of them are not included in this study despite the fact that some of them have been performed in China.

Response 6: Thank you for your valuable comments. We increase the research results in China as follows (Lines 79-84):

“[16] and [17] analyzed the characteristics of tunnel construction accidents in China in terms of temporal distribution, spatial distribution, accident levels and accident types us-ing statistical analysis methods. [18] also counted 10 years of tunnel accident incident cases and used the N-K model to construct a coupled model of tunnel construction risk to reveal the coupling effect between multiple risk factors of tunnel construction accidents.”

  1. Zhu, Y.; Zhou, J.; Zhang, B.; Wang, H.; Huang, M. Statistical analysis of major tunnel construction accidents in China from 2010 to 2020. Tunnelling and Underground Space Technology 2022, 124, 104460.
  2. Zhang, J.; Chen, Y.; Chen, T.; Mei, Z. Law and characteristics analysis of domestic tunnel construction accidents from 2006 to 2016. Modern Tunnelling Technology 2018, 55, 10-17.
  3. Hongwei, P.; Desai, G.; Zhanping, S.; Tian, X.; Yuwei, Z.; Libo, D. Coupling Analysis to Investigate Multiple Risk Factors for Tunnel Construction Accidents Based on N-K Model. Tunnel Construction 2022, 42(9), 1537-1545.

Round 2

Reviewer 1 Report

Thank you for the revision of the paper and the answers.

Some of the answers provide useful clarification. However, we do not find all these details in the article. Moreover, the modification of the discussion does not really change the contribution of the paper, see my position in point 21 below.

Point 2/Point 3 : Give details in the abstract and the title on the subject of the study: Road tunnel accident, China.

Point 12 : Figure 1 should be modified to reflect the fact that DEMATEL model follows the Grounded theory model. Here, they are in parallel.

Point 13 : Once again, the paper lacks information on the representativeness of these 30 accidents and the selection choices that were made. For example, what keywords (or other method) were used to extract these 30 reports.

Point 15 : The comment was rather aimed at the following sentence : “Referring to the grounded theory saturation principle[26], the ratio of sampling coding samples and theoretical saturation test samples is set to 2:1; that is, 20 coding samples and 10 test samples.” To be put in 2.4.4.

Point 16: No change in the paper.

Point 18 : Add these explanations in the article.

Point 20 : Personally, I would have added the b(i) in table 4 for the benefit of the reader.

Point 21 : There are improvements in the discussion, but the essence of my comment has not really been answered. Indeed, the discussion of the study does not add much new information in its current form. For example, factors such as "safety management", "regional climatic environment" or "engineering hydrogeology" are pointed out without going into detail. The added value of the qualitative approach of this study, compared to other studies, does not emerge in the discussion in my opinion. There should be concrete explanations by coming back with examples from the accidents.

I have no problem with the article being published in this form, but I find that we are somewhat missing its potential contribution, which is not the methodological aspect, but the fact of being able return to link the influencing factors to concrete examples. 

Author Response

Response to Reviewer 1 Comments

Point 1: Give details in the abstract and the title on the subject of the study: Road tunnel accident, China.

Response 1: Thank you for your valuable comments. We have modified the title as follows:

"Factors Affecting Road Tunnel Construction Accident in China Based on Grounded Theory and DEMATEL"

Details have also been added to the abstract(Lines 11-12):

“Despite the continuous progress of tunnel construction technology and safety management technology, road tunnel construction safety still faces many challenges in china, ”

Point 2: Point 12 : Figure 1 should be modified to reflect the fact that DEMATEL model follows the Grounded theory model. Here, they are in parallel.

Response 2: Thank you for your valuable comments. We have modified Figure 1 as follows(Line 147):

Point 3: Once again, the paper lacks information on the representativeness of these 30 accidents and the selection choices that were made. For example, what keywords (or other method) were used to extract these 30 reports.

Response 3: Thank you for your valuable comments. We have added the methodology for selecting reports as follows(Lines 179-185):

"Search and collect accident reports on the official websites of various government departments using keywords such as "tunnel", "construction", and "accident report". Then the accident reports were statistically analyzed, including the geographical location of the accident, the number of casualties, the size of the tunnel and other factors, and selected representative classic accident reports, including 30 official accident investigation reports from 2011-2021 from government production safety supervision and management departments at all levels."

Point 4: The comment was rather aimed at the following sentence:” Referring to the grounded theory saturation principle[26], the ratio of sampling coding samples and theoretical saturation test samples is set to 2:1; that is, 20 coding samples and 10 test samples.”

Response 4: Thank you for your valuable comments. However, the coding step as a whole is divided into three steps, open coding, axial coding, and selective coding. According to the principle of sample selection, we first use 20 samples for coding, and then use the remaining 10 samples for saturation testing. The saturation test steps are also performed in the order of open coding, axial coding, and selective coding. Therefore, we thought it would be clearer to write 20 samples directly for coding and the remaining 10 for saturation testing before starting coding. We also show in 2.4.4 that the samples for the saturation test are the remaining 10. We also added a little detail as follows(Lines 243-245):

"Two coders independently encoded the tunnel construction accident influencing factors as the core, and used NVivo11 software to analyze 20 the tunnel construction accident case coding sample materials sentence by sentence. "

Point 5: Point 16: No change in the paper.

Response 5: Thank you for your valuable comments. We have modified this vulnerability.

Point 6: Add these explanations in the article.

Response 6: Thank you for your valuable comments.  We added the explanationas follows (Lines 374-377):

"The invited experts are senior engineers with extensive experience in tunnel construction. We used the purpose sampling to select five experts, and then through the snowball sampling, looking for more experts, and finally got 20 samples."

Point 7:  Personally, I would have added the b(i) in table 4 for the benefit of the reader.

Response 7: Thank you for your valuable comments. We have added bi in Table 4.

Point 8: There are improvements in the discussion, but the essence of my comment has not really been answered. Indeed, the discussion of the study does not add much new information in its current form. For example, factors such as "safety management", "regional climatic environment" or "engineering hydrogeology" are pointed out without going into detail. The added value of the qualitative approach of this study, compared to other studies, does not emerge in the discussion in my opinion. There should be concrete explanations by coming back with examples from the accidents.

Response 8: Thank you for your valuable comments. We We have added the relevant case descriptions from the accident reports as follows (Lines 426-438, 464-470, 502-505, 523-531) :

“In the accident reports, there are problems of inadequate personnel management at the construction site, such as "At the time of the incident, the personnel of the shield machine in the left line for opening and changing the knife did not match with the personnel listed in the special plan, and the change of personnel was not reported to the supervisory unit for review and approval."; there are mechanical equipment inspection and maintenance safety management is not in place, such as "not mobile and manual power tools use, repair, maintenance inspection, resulting in angle grinder missing protective cover problem was not found in time to solve, does not meet the requirements of use was used for site construction.""

“such as “It is inferred that the coupling effect of various unfavorable geological conditions and climatic reasons such as earthquake and sufficient rainfall into April is the main rea-son for the overall brittle damage and instantaneous and sudden local collapse of the vault." and  "After the accident, the accident investigation team expert group after site in-vestigation, found that the geological conditions are actually striped mixed granite, lamel-lar development, the existence of kaolin and other unfavorable geological conditions of water softening."“

“As in the accident reports, "The direct cause of this accident is: the collapse of the lower section of the formwork due to a combination of defects in the formwork welds, missing flange connection bolts, and excessive speed of concrete pouring.""

“In the construction accident reports, "did not organize the relevant personnel of the unit to carry out highway engineering safety supervision and inspection training, resulting in the low quality of safety supervisors, can not effectively perform their supervisory duties."; "The pump truck operator, with unclear vision and without the command of the pump truck signalman, manually operated the pump truck in violation of the law, which directly led to the accident.", " tunnel construction team safety technical briefing, daily safety education and training system are not implemented, construction personnel on the construction site safety risk awareness is not in place, safety awareness is weak, poor self-safety prevention, is an important cause of the accident.””

Point 9:  I have no problem with the article being published in this form, but I find that we are somewhat missing its potential contribution, which is not the methodological aspect, but the fact of being able return to link the influencing factors to concrete examples. 

Response 9: Thank you for your affirmation! We have added relevant examples.
